# Environmental Pollutants, Mucosal Barriers, and Pathogen Susceptibility; The Case for Aflatoxin B_1_ as a Risk Factor for HIV Transmission and Pathogenesis

**DOI:** 10.3390/pathogens10101229

**Published:** 2021-09-23

**Authors:** Erin P. Madeen, Frank Maldarelli, John D. Groopman

**Affiliations:** 1Department of Cancer Prevention, National Institute of Health, Shady Grove, MD 21773, USA; 2HIV Dynamics and Replication Program, NCI-Frederick, Frederick, MD 21703, USA; fmalli@mail.nih.gov; 3Environmental Health and Engineering, Bloomberg School of Public Health, Johns Hopkins University, Baltimore, MD 21205, USA; jgroopm1@jhu.edu

**Keywords:** aflatoxin, mycotoxins, HIV, inflammation, environmental pollution, mucosal barrier, cytokines, chemokines, susceptibility

## Abstract

HIV transmission risk is dependent on the infectivity of the HIV+ partner and personal susceptibility risk factors of the HIV− partner. The mucosal barrier, as the internal gatekeeper between environment and self, concentrates and modulates the internalization of ingested pathogens and pollutants. In this review, we summarize the localized effects of HIV and dietary toxin aflatoxin B1 (AFB_1_), a common pollutant in high HIV burden regions, e.g., at the mucosal barrier, and evidence for pollutant-viral interactions. We compiled literature on HIV and AFB_1_ geographic occurrences, mechanisms of action, related co-exposures, personal risk factors, and HIV key determinants of health. AFB_1_ exposure and HIV sexual transmission hotspots geographically co-localize in many low-income countries. AFB_1_ distributes to sexual mucosal tissues generating inflammation, microbiome changes and a reduction of mucosal barrier integrity, effects that are risk factors for increasing HIV susceptibility. AFB_1_ exposure has a positive correlation to HIV viral load, a risk factor for increasing the infectivity of the HIV+ partner. The AFB_1_ exposure and metabolism generates inflammation that recruits HIV susceptible cells and generates chemokine/cytokine activation in tissues exposed to HIV. Although circumstantial, the available evidence makes a compelling case for studies of AFB_1_ exposure as a risk factor for HIV transmission, and a modifiable new component for combination HIV prevention efforts.

## 1. Introduction

The mucosal surface acts as the initial barrier to HIV acquisition, preventing interactions with underlying epithelial tissues and immunological cells. The most vulnerable sites for sexual transmission (mucosal barrier translocation) of HIV, in order of highest to lowest susceptibility, are colorectal/anal tissues, vaginal tissues, foreskin, and penile/urethra tissues (reviewed in [1,2]). Factors such as the physical anatomy of the barrier, cellular immunity, soluble factors, and interactions between the epithelial barrier and the local microenvironment, such as mucus and host microbiota can block viral entry [3,4]. Defects to this barrier are a risk factor for microbial infections and HIV acquisition is elevated in individuals with mucosal inflammation, such as those with sexually transmitted infections (STIs) Environmental agents, such as aflatoxin B1, are immunotoxicants have been described in veterinary and toxologic studies, but their potential role as inflammatory agents acting at mucosal sites and contributing to HIV transmission and pathogenesis remains largely unexplored. To shed light on the potential role of immunotoxicants as cofactors in HIV transmission and pathogenesis, we review the mucosal immunotoxic characteristics of environmental pollutants, such as aflatoxin B1, in the context of the local settings where HIV and toxin exposure is prevalent.

Immunotoxicants represent significant health challenges in low and middle income countries including those with high HIV prevalence and incidence. HIV is epidemic in many low- and middle-income countries with high environmental pollution [5], including countries such as: Botswana, Lesotho, South Africa, Swaziland, Mozambique, Namibia, Zimbabwe, Zambia, and Malawi, reviewed in [6]. South Africa has the world’s largest HIV-positive population (7.7 M people, 20.4% of residents) [7]; countries with the highest prevalence of HIV also have high environmental exposures. Lower income countries have higher rates and greater variability in HIV transmission per sexual act than higher income countries when normalized and corrected for known co-factors, suggesting geo-local, unaccounted risk factors [8]. The dietary environmental pollutant and International Agency for Research on Cancer (IARC) class-1 carcinogen, aflatoxin B1 (AFB_1_), is prevalent in many regions with high HIV instances, including China and low and middle income countries in Sub-Saharan Africa and South East Asia [9,10]. AFB_1_ is a known immunomodulator, altering the translocation of immune cells, initiating inflammation, and acting as an immunosuppressant [11]. In addition to social and health care related risk factors for HIV transmission, we hypothesize that environmental pollutant exposures, such AFB_1_ ingestion, are risk factors for the susceptibility and infectivity of sexually acquired HIV.

AFB_1_ is primarily studied as a potent hepatocellular carcinogen, however it is also linked to negative health endpoints in individuals with one of several viral infections. It leads to synergistic development of hepatocellular carcinoma, morbidity, and death during hepatitis B virus and/or hepatitis C virus co-infection, increases influenza viral replication, and is associated with oncogenic human papilloma virus [12,13,14]. Most importantly, AFB_1_ exposure is correlated with higher HIV blood plasma viral load and is being investigated as a risk factor for HIV progression [15,16]. AFB_1_ has yet to be studied as a risk factor for HIV acquisition, high viral set point, or isolated HIV shedding in semen or vaginal secretions. In addition, as immune activation is linked to morbidity and mortality during suppressive long-term antiretroviral therapy (ART), AFB_1_ -induced immune activation may represent a risk factor for disease progression during therapy [17].

AFB_1_ is a mycotoxin produced by *Aspergillus sps.* on staple crops including corn and peanuts and is difficult to control without storage and monitoring programs. AFB_1_ can grow on crops in the field during environmental distress, such as drought or pest damage. It can also grow on harvested crops stored in humid or improper conditions [18]. Cooking or rinsing does not remove AFB_1_ due to the molecular stability of aflatoxins (pH and thermal stability, weakly soluble in aqueous solvents) [19,20]. Stability is likely attributable to the polycyclic structure with several sites of unsaturation. Identifying and removing individually contaminated kernels or discarding a contaminated batch is the usual route to limit human exposure [18]. In resource limited contexts and during subsistence farming, monitoring programs for AFB_1_ contamination are rare, generating higher rates of AFB_1_ exposure in low socioeconomic regions.

AFB_1_ exposure is highest in many regions with high HIV prevalence. Sub-Saharan Africa, including South Africa, has the highest exposures with up to 400 ng AFB_1_/kg body weight/day [21]. Low income countries have a typical exposure range of 0.1 to 49 ng AFB_1_/kg body weight/day while high income countries rarely have exposures above 1 ng AFB_1_/kg bodyweight/day see [21] for review. AFB_1_ is a pro-toxin. Enzymatic activation is required to convert inert AFB_1_ to a reactive toxic metabolite, AFB_1_ exo-8,9-epoxide, capable of reactive oxygen species generation, inflammation induction and genotoxicity [22]. Yet, other oxidative AFB_1_ metabolites also redox cycle [23]. AFB_1_ is primarily metabolically activated by cytochrome P450 3A4 [22]. AFB_1_ localization in colorectal and cervicovaginal tissues could provide an important link between environmental factors and HIV transmission, as inflammation can disrupt mucosal barriers, locally recruiting HIV susceptible cells to the tissues of exposure. AFB_1_ tissue localization at sites of sexual HIV transmission can occur by three routes: systemic distribution from an oral ingestion, delivery via sexual fluids containing AFB_1_, or infection with *Aspergillus flavus* (the fungus that produces aflatoxins)reviewed in [24,25].

AFB_1_ is often studied in the context of liver toxicity, however the GI tract is the site of absorption from an oral exposure, prior to distribution to the liver through the portal vein. Various animal models have shown that AFB_1_ is absorbed by passive transport primarily in the upper GI (80% of administered dose) [26,27]. Remaining unabsorbed AFB_1_ is available to the lower GI lumen. Disruption of peripheral blood based immunological markers are associated with increasing AFB_1_ exposure in people from high AFB_1_ exposure areas [11,28,29]. Rodent AFB_1_ exposure models have altered intestinal morphology, including an increase in villus length, width and crypt depth and a decrease in mucus-producing goblet cell number, and a decrease in microflora genera and phyla representation [30,31].

## 2. Effects of AFB_1_ Exposure on HIV Susceptibility

### 2.1. HIV Transmission and AFB_1_ Localization in Distinct Anatomic Tissues

#### 2.1.1. Cervicovaginal HIV Transmission and Localization of AFB_1_

The integrity of the vaginal mucosal barrier is a critical local mechanism to prevent the transmission of HIV. Cervicovaginal tissues are susceptible to HIV transmission and infection in a location specific manner, founder HIV infection foci are found primarily in the endocervix and zone of endo- and ectocervical transition [32]. The endocervix and transformation zone contains only a single layer of columnar epithelium, which is more permissive to viral translocation than vaginal tissue comprised of stratified squamous epithelium [33]. A healthy cervicovaginal mucosal barrier traps and eliminates the vast majority of HIV virions, preventing widespread founder infection throughout the reproductive tract exposed to HIV [32,34]. Decreased integrity of the cervicovaginal mucosal barrier through inflammation or mechanical action provides an opportunity for microbial translocation to susceptible cells in the cervicovaginal tract.

AFB_1_-DNA adducts are formed in cervicovaginal tissue, demonstrating that AFB_1_ does reach cervicovaginal tissue and that it is activated to the reactive epoxide in or near the cervix [35]. Vaginal tissue expresses CYP P450 3A4, while cervical expression is minimal [36,37]. Additionally, vaginally delivered AFB_1_ is absorbed across the mucosal membrane and into circulating blood plasma with a T_max_ of 30 min in dairy cattle [38]. AFB_1_ localizes to semen following dietary exposure [39]. It is logical that AFB_1_, delivered via ejaculate, will cross the mucosal tissue in humans to localize near target immunological cells in the cervicovaginal or colorectal sites of HIV infection. Oral or sexual exposure and distribution localizes reactive AFB_1_ to the cervicovaginal tissue susceptible to HIV during vaginal sexual exposure.

#### 2.1.2. Vaginal Inflammation and HIV Transmission

Inflammation in cervicovaginal tissues generates local recruitment CD4^+^ T cells, increasing seroconversion risk by increasing the population of susceptible cells [33]. Vaginal mucosal barrier damage can result in increased susceptibility to HIV as well as increased pathogenesis, morbidities, or mortality in established HIV infection [3]. Dendritic cells are the first to take up HIV-1 and to present it to T cells, where it replicates in vaginal and ectocervical mucosa [40].

Local cervicovaginal inflammation increases the risk of HIV seroconversion and increases genital viral shedding, increasing susceptibility and infectivity [41]. Inflammation can be generated by a variety of mechanisms, including co-morbid infections or other agent exposure, that generate a variety of inflammatory signaling cascades. Women who have elevated levels of at least 5 pro-inflammatory cytokines in vaginal mucosa, particularly MIP-1a, IL-8, MIP-1b, IL-1b, IL-1 α, and TNF- α, are three times more likely to acquire HIV (Table 1) [3,42]. An increased vaginal pro-inflammatory cytokine profile is associated with mucosal barrier disruption and an increased CD4^+^ T cell concentration, increasing the risk of HIV translocation and infection [43]. Additionally, increased cervicovaginal expression of chemokine IP-10 generates increased cervicovaginal HIV viral shedding [41]. While AFB_1_ is a known inflammatory toxicant, and its metabolites have been detected in the cervicovaginal tract, the inflammatory effects resulting from AFB_1_ localization to cervicovaginal tissue remain unknown.

HIV infection risk is greatest to women in low income countries. HIV infection rates are eight times higher among teenage women than among teenage men in South Africa [7,62,63]. HIV prevalence is 5.6% in 15–19-year-old women and is 17.4% in 20–24-year-old women (11.8% increase), relative to a 4.4% increase (0.7% to 5.1%) in men between the same ages [7,63]. Per-sex-act risk of HIV transmission in low income countries is 3.75-fold higher from men to women and 9.5-fold higher from women to men than in high income countries [8]. The regional higher HIV transmission rates and especially high risk to women are speculated to be attributable to high rates of sexual violence against women, low rates of HIV prevention knowledge, low circumcision rates, and increased HIV susceptibility from cervicovaginal inflammation induced by BV, STI, forced sex, and use of vaginal drying agents (believed to enhance the male partner’s sexual pleasure) [64]. Cervicovaginal inflammation induced mucosal barrier disruption is poorly understood from any of the above factors, let alone from pollutant exposure, in the context of HIV susceptibility.

#### 2.1.3. STI and HIV

Sexually transmitted infections (STI) are the most common sources of genital inflammation studied in the context of HIV, and STI co-infection generated inflammation increases HIV replication and susceptibility. The presence of STIs, such as syphilis, gonorrhea, chlamydia, and herpes simplex virus facilitates HIV transmission, as reviewed in [65,66]. Mechanisms of increased transmission risk include breakdown of mucosal barrier andincreased presence of activated immune cells. Even in STIs where mucosal barriers are not always compromised, such as gonorrhea, there are increases in immune cell infiltration, with increased HIV transmission risk [66]. Other viral infections represent important HIV transmission risks as well. In a 20-year cohort study from Kenya, prevalent HSV-2 is the largest contributor to HIV acquisition risk (48.3%) out of the metrics measured [67]. The proposed mechanisms of HSV-2 driven HIV infection risk are inflammation, ulceration, localized T-cell recruitment, and increased expression of CCR5 [68]. These endpoints should also be investigated following exposure to xenobiotics that localize to sexual tissues, such as AFB_1_, to determine the effects on HIV risk.

Microbiome alterations as an HIV risk factor is an emerging topic [69]. It is better understood in vaginal tissues than in colorectal tissues, due to the relative simplicity of the vaginal microbiome. The vaginal microbiome is primarily a *Lactobacillus* monoculture. BV is defined as vaginal primary colonization by microbe species other than *Lactobacillus* and it is associated with an increased risk of HIV infection [70]. However, only 37% of healthy, HIV-, young women in a South African study are predominately colonized by *Lactobacillus* [71]. High vaginal bacterial diversity correlates with high expression of inflammatory cytokines and women with elevated pro-inflammatory genital cytokines have more activated cervical HIV target cells (Table 1).

#### 2.1.4. Colorectal HIV Transmission

The colon is a critically important tissue as the route of highest risk of HIV transmission; HIV risk is 17-fold greater in unprotected receptive anal intercourse (URAI) than in unprotected receptive vaginal intercourse [1]. Additionally, an insertive HIV negative partner has a higher HIV risk per-sex-act during anal sex than during vaginal sex [1]. Often URAI studies are in men who have sex with men (MSM) and resulting HIV transmission. However, URAI within heterosexual relationships is a common and underreported practice [72]. As of 1999, the absolute number of self-reporting women in the USA practicing URAI was ~7-fold greater than the number of MSM practicing URAI [73]. Heterosexual URAI as a means of HIV transmission is an important factor in HIV epidemics as it links a high-risk behavior to a greater percentage of the population [74]. In the United States, receptive anal sex is attributable to 40% of acquired HIV in heterosexual women ages 18–34, while insertive anal sex is attributable to 20% of acquired HIV in heterosexual men [75]. Geographically, this becomes important relative to co-exposure with colorectal immunotoxicants, such as AFB_1_, as a risk factor for URAI acquired HIV. Over 75% of female sex workers participating in a South African site of a multi-center microbicides trial reported URAI during follow-up [74,76], indicating that the colorectal tissue route of HIV exposure is relevant in regions with high levels of AFB_1_ exposure.

#### 2.1.5. Colorectal Localization of AFB_1_ and Pathology

AFB_1_-DNA adducts have been detected in rectal and lower colon tissues in cadavers from The United Kingdom, with a trend for higher concentration in tumor tissue relative to non-tumor tissue [35]. AFB_1_ activating cytochrome P450 3A4, is consistently expressed as protein in the ascending, descending, and sigmoidal colon mucosa [22,77]. The pathogenic role of AFB_1_ in the lower GI tract has not been studied in humans. However, various endogenous and microbial synthases can initiate AFB_1_ redox cycling, initiating inflammation pathways outside of the typically studied cytochrome P450 activated epoxide generated genotoxicity [78,79,80]. The expression of AFB_1_ metabolizing enzymes and AFB_1_-DNA adducts in colorectal tissues and tumors indicates that AFB_1_ metabolism and inflammatory effects warrants further research in lower GI tissue.

#### 2.1.6. HIV and GI Barrier Function and Immunity

Gut associated lymphoid tissue contains 80% of the total body lymphocyte pool in healthy individuals, while peripheral T lymphocytes only represent 2–5% of total lymphocytes [81], making the gut a key tissue in HIV infection and progression. Little is known about AFB_1_ specific inflammation mechanisms in the human GI, especially the lower colorectal tissues exposed to HIV during receptive anal sex. Animal studies have identified AFB_1_ induced alterations in histology [82], oxidative stress [83], mucous composition, and barrier function of the gut [84], as well changes in the composition of the microbiome [85,86]. Substantial effects of AFB_1_ exposure on transcriptome profiles [87] and on T cell subsets have been reported in poultry (reviewed in Fouad et al. [88]. We are unaware of how AFB_1_ exposure affects the gut associated lymphoid tissue in humans or modulates susceptibility to HIV infection. Mucosal barrier disruption increases the potential for xenobiotic and microbial translocation into tissues, as a healthy mucosal barrier reduces the bioavailability of AFB_1_ by binding and eliminating AFB_1_ prior to interactions with the GI epithelium [89].

Investigating known GI inflammatory pathways for parallel modulation by common pollutants, including AFB_1_, may provide insight into a local environmental risk factor for HIV susceptibility, increased viral load or set point, and pathogenesis. HIV infection activates inflammation pathways that are well described and are logical mechanisms to explore in the context of co-exposure and co-morbidities that generate GI immunotoxicity. HIV induces inflammation through stimulation of the innate and adaptive immune systems [90] that increases HIV progression, morbidity, and mortality [91]. Depletion of the CD4^+^ T helper cells 17 (Th17) cell population associated with the GI tract [92] is a hallmark of HIV infection; Th17 cells are responsible for protection against extracellular pathogens, and maintain mucosal barrier integrity [93]. During untreated HIV infection, Th17 cell populations decrease, leading to GI immune activation and inflammation, which is only partly restored by antiretroviral therapy [94], and results in mucosal barrier disruption with localized inflammation due to microflora microbial translocation into the GI epithelium, which has widespread systemic effects with cytokine disruption (Figure 1) [52,58,95,96].

Detailed studies of immune alterations after AFB_1_ exposure have not conducted, although in cytokine profiles have been reported in animal studies (Table 1). AFB_1_-induced upregulation of IL-1α, IL-4, IL-6, IL-8, TNF-alpha, and Ifn gamma have been reported, all of which are reported risk factors for increased HIV transmission (Table 1). Other cytokines (e.g., IL-2) are not upregulated by AFB_1_, and further research will be essential in determining the role of AFB_1_-induced immune activation in HIV transmission. Mucosal barrier disruption increases the potential for xenobiotic and microbial translocation into tissues, as a healthy mucosal reduces the bioavailability of AFB_1_ by binding and eliminating AFB_1_ prior to interactions with the GI epithelium [89].

Colon specific markers and samples must be considered for future studies to answer accurately address cellular recruitment and inflammation in the colorectal compartment. Studies of peripheral CD4^+^ cells do not reflect CD4^+^ conditions in the colon. GI CD4^+^ cell counts do not correlate with circulating CD4^+^ cell counts, and total CD4^+^ cell counts do not factor in functionality or cytokine/chemokine expression by those cells [97,98]. To better describe the colon specific CD4^+^ cell types and the effects of inflammation on colon viral susceptibility, animal models and biopsy studies may be necessary with a combination of pathology and flow cytometry tools for cell typing.

### 2.2. AFB_1_ Exposure and HIV Viral Load

#### 2.2.1. HIV Copy Number and Transmission Risk

The transmission risk for HIV is attributable to two factors, the infectivity of the HIV+ partner, and the susceptibility of the HIV− partner. HIV replicates in T cells, and migrates via macrophages and dendritic cells, thus transmission is affected by the immunocompetency of either or both partners. Blood plasma viral load (BPVL) is generally correlated to semen viral load (SVL), with a lower viral RNA copy number in semen relative to blood plasma [99].

Anti-retroviral therapy (ART) for “treatment as prevention” is highly successful in decreasing HIV transmission by decreasing the HIV+ partner’s BPVL to below the threshold associated with genital fluid viral loads necessary to establish a sexual infection in a HIV− partner [100]. HIV serodiscordant couples who were not using condoms for vaginal or anal intercourse were monitored over 30 months with the following results: the seroconversion rate was 22% for couples with a blood plasma viral load higher than 50,000 copies/mL, but dropped to 5% seroconversion between 400 to 4000 copies/mL, meanwhile, no transmissions occurred in couples with a viral load below 400 copies/mL [101,102]. The PARTNER2 study demonstrated that there is no risk of HIV transmission in serodiscordant European MSM couples having URAI if the HIV BPVL is maintained below 200 copies/mL with antiretroviral therapy [100], leading to the “undetectable = untransmissible” campaign abbreviated “U = U”. It should be noted that this was based on a European cohort with access to ART, while stigma, supply chain, and cost related issues prohibit widespread ART access and compliance in many low income, HIV epidemic regions. Additionally, several ART drugs, including protease inhibitors, are known to inhibit cytochrome P450 3A4, the enzyme that metabolizes AFB_1_ to the reactive AFB_1_ exo-8,9-epoxide [22,103]. In addition to reducing HIV BPVL and/or SVL, ART may decrease AFB_1_ epoxide concentrations, shunting the AFB_1_ parent to less inflammatory metabolism and elimination pathways, reducing AFB_1_ induced inflammation.

Discordance between HIV BPVL and SVL is a noted phenomenon, referred to as “isolated HIV semen shedding”, that lacks an etiological mechanism though elevated/detectable SVL during undetectable BPVL is a potential HIV transmission risk. The general trend of undetectable SVL in individuals with undetectable BPVL is the basis for utilizing blood sample based assays to determine viral load and resultant infectivity [104]. Deviation from that trend, discordance between BPVL and SVL, has been noted in multiple studies [99,105,106,107]. SVL is increased by local genital tract inflammation [99,108]. HIV replication generates genital tract inflammation, resulting in a positive feedback loop increasing HIV replication in the genital tract and recruitment of HIV susceptible cells [109,110]. Additionally, urogenital inflammation, defined via leukocytospermia, without STI co-infection is common during SVL/BPVL discordance. The cause of increased SVL associated leukocytospermia is idiopathic, but recent unprotected insertive anal sex is a hypothesized risk factor [105].

Inflammatory xenobiotics that localize to the male genital compartment, such as AFB_1_, may increase the SVL relative to BPVL, generating isolated HIV semen shedding in a similar mechanism to STI co-infection. The male reproductive tract is a target for AFB_1_ localization and AFB_1_ is shed in semen. AFB_1_ concentrations in boar feed have a positive correlation with AFB_1_ concentrations in boar semen [111]. In regions with dietary AFB_1_ exposure, human semen contains AFB_1_ [39]. It is likely that human semen AFB_1_ levels also increase with increasing AFB_1_ dietary exposure [112]. High AFB_1_ exposure in HIV+ individuals may contribute to genital tract inflammation and increased seminal viral load, which would be a HIV seroconversion risk to a HIV− partner.

#### 2.2.2. AFB_1_ Exposure and HIV Copy Number

AFB_1_ exposure in HIV+ individuals has a positive correlation with viral copy number and copy number correlates to temporal trends in aflatoxin ingestion in Ghanaian retrospective studies [11,15,16]. The blood plasma HIV viral load in individuals with high aflatoxin exposure (aflatoxin-albumin adduct > 0.93 pmol mg^−1^ albumin) was three-fold higher than that of low AFB_1_ exposure individuals (aflatoxin-albumen adduct < 0.93 pmol mg^−1^ albumin) [16]. Blood plasma aflatoxin-albumin adduct concentration is a remarkably stable biomarker representing accumulated adducts during the prior 30 days (approximate longevity of albumin protein) [113]. The proposed mechanism of AFB_1_ exposure driven increases in HIV copy number is an increase in HIV transcription [16]. In porcine models of immunization during AFB_1_ exposure, AFB_1_ prevents antigenic recognition and specific immunity, but does not prevent T cell proliferation [47]. AFB_1_ exposure hampers adaptive immunity, leading to reduced immunization efficacy and defenses against infectious disease [47]. Potentially, aflatoxin exposure is a risk factor for HIV transmission and progression through an increase in viral copy number in the HIV+ partner.

The HIV viral load set point is the stabilized blood plasma viral load in the period following the initial acute HIV infection period prior to ART initiation, which is positively correlated with HIV progression. The increasing HIV viral load with increasing AFB_1_ blood plasma albumin adducts indicates that AFB_1_ chronic exposure may also increase the viral load set point. The determinants of HIV viral set point remain unknown, but pollutant exposure has yet to be investigated as a factor [16,114]. The viral setpoint peaks two weeks after HIV infection and reaches stability four weeks post infection [115]. This is important relative to pollutant exposure because pollutant exposure may increase immune activation and the local population of susceptible cells through local inflammation. Thus, inflammation status at early infection time points may affect the viral setpoint, which in turn affect the HIV+ cell reservoir. The higher the set point and the longer time to ART initiation/compliance results in a higher HIV+ cell reservoir, associated with quicker viral load rebound after ART cessation and greater HIV associated morbidity [115]. Thus, if AFB_1_ generates pre-existing systemic inflammation, it may increase the HIV viral set point during initial HIV infection, which increases HIV progression in unmanaged HIV and increases HIV infectivity.

### 2.3. HIV Progression and Inflammation

#### 2.3.1. Inflammation and HIV Replication/Progression

Inflammation factors recruit and activate HIV susceptible cells at sites of HIV exposure and are implicated in HIV transmission (Table 1). Elevated levels of α-defensins at mucosal surfaces are associated with an increased risk of becoming HIV infected [3,54,55]. High levels of HIV replication are associated with high concentrations of pro-inflammatory cytokines, including IL-6, TNF-α, IFN-Υ, IL-18, and IL-12 (Table 1). Regulation can be multidirectional. HIV replication in resting CD4^+^ T cells isolated from virally suppressed individuals can be reactivated by IL-6, IL-2 and TNF- α [49,52] and HIV replication induces the expression of IL-6 and TNF-α in intestinal cell models [51]. Mucosa with high levels of viral replication contain high levels of IFN-γ in natural killer cells and T cells. IFN-γ drives the differentiation of CD4^+^ cells, providing local cellular targets for HIV infection and sites of viral replication [46,52]. Increased local inflammation at mucosal barriers increases susceptibility to HIV infection, and HIV replication induces mucosal barrier inflammation. This cycle of inflammation and microbial translocation is vulnerable to modulation by environmental factors, such as pollutants and microbes and it merits further investigation at a tissue and agent specific level.

#### 2.3.2. Cytokines Associated with HIV

The latent HIV reservoir that persists following ART initiation is a target for a HIV cure. Pro-inflammatory cytokine upregulation is a proposed mechanism maintaining the latent reservoir through the production of resting CD4^+^ cells [52]. Successful antiretroviral therapy is linked to a decrease in blood serum IL-18 levels, while treatment failure is often associated with elevated IL-18 levels, indicating that IL-18 production may increase viral replication [52,60]. IL-18, produced primarily by monocytes/macrophages, enhances HIV replication in monocyte and T cell lines. Resting CD4^+^ T cells are more susceptible to HIV infection when co-exposed to gamma chain (ϒC) cytokines, including IL-2, IL-4, IL-7, and IL-15, (Table 1) likely by encouraging cell cycle progression to the G1b phase [52,53]. Latent resting CD4^+^ T cells, in the absence of ART, are reactivated by exposure to ϒC cytokines, increasing HIV viral load. ϒC cytokines have been investigated as “shock and kill” HIV immunotherapy for their ability to reactivate the latent resting CD4^+^ T cells in an attempt at an HIV cure [52,116]. In depth studies on the effects of AFB_1_, or other environmental pollutants, on cytokine/chemokine expression are limited, though AFB_1_ is recognized as an immunotoxicant [117,118]. Long-lived HIV-infected cells may undergo clonal expansion in response to homeostatic mechanisms or antigen stimulation. As an immunomodulatory agent, AFB_1_ may stimulate expansion of T cells, including HIV-infected T cells, and exert a direct effect on the HIV reservoir.

## 3. AFB_1_ and HIV Co-Exposure Research Design Considerations

### 3.1. Exposure Design

Aflatoxin and HIV exposure overlap in several significant geographic, mechanical, tissue-local, and at-risk population level metrics described in this review. To investigate the potential impact of AFB_1_ dietary exposure as a risk factor increased HIV susceptibility, basic metabolic data is necessary for appropriate research design. A large barrier and the first obstacle to designing appropriate experiments testing the effects AFB_1_ exposure in HIV seroconversion risk is the lack of data describing aflatoxin administration, distribution, metabolism, and elimination (ADME) in humans. There is evidence that AFB_1_ localizes to sexual tissues [35,39], however the tissue specific concentration and metabolic state of aflatoxin after dietary or sexual fluid exposure is largely unknown. Biopsy and ex vivo experiments of human cervicovaginal and anorectal tissues are appropriate models for the local AFB_1_ metabolic activation and subsequent pathological and immunological effects of AFB_1_ exposure as it relates to known HIV risk factors such as cellular recruitment, immunotoxicity, mucosal barrier integrity, and cellular junction “leakiness” [3,4,119,120]. Additionally, cervicovaginal and anorectal biopsy, cytology collections, or lavage of people with high AFB_1_-blood serum lysine adducts, and controls can generate tissue local immunological response data.

The second major hurdle in viral and environmental pollutant co-exposure models is appropriate in vivo model selection. As AFB_1_ is an International Agency for Research on Cancer (IARC) Group-1 carcinogen, human clinical trials of ADME following dietary exposure are not appropriate [9]. Yet a whole system model of dietary ingestion, ADME, and sexual tissue local effects are necessary due to the complexity of xenobiotic and viral co-exposure. Analogues of human CYP450 3A4, responsible for metabolizing pro-toxicant AFB_1_ to the reactive metabolite (AFB_1_-exo-8,9-epoxide), must be expressed in the system model, as well as the appropriate constitutive expression of glutathione-S-transferase A3 (GST-A3) analogues for the deactivation and elimination of AFB_1_ metabolites [22,29]. Wild-type mice are excluded as a model system due to high constitutive expression of GST-A3, reducing their sensitivity to AFB_1_ toxicity [121]. Rhesus macaques and simian immunodeficiency virus (SIV) are the gold standard non-human primate model of human HIV infection, but rhesus macaques have high AFB-glutathione conjugating activity relative to humans, decreasing their sensitivity to AFB_1_ exposure [122]. Rats have been the representative experimental model for human aflatoxin metabolism and recently rat models of human HIV infection and prevention have become available [121,123]. Due to the large body of literature in rat models of AFB_1_ exposure and the low research cost of a rodent model relative to a non-human primate model, the rat model may be the most appropriate model for basic research on the ADME of oral AFB_1_ exposure as it relates to HIV susceptibility.

Dosing and duration are critical in AFB_1_ exposure selection as human dietary exposure is chronic, low dose, and sub-acutely toxic. Human dietary exposures range between 400 ng AFB_1_/kg body weight/day and less than 1 ng AFB_1_/kg bodyweight/day [21]. A range finding study in rats utilized an oral dosage range of 2.2, 73, 2110 ng/kg per day for between 4–8 weeks, with the higher dosage range representing a carcinogenic dosage, led to detectable (via radiolabel and scintillation counting) AFB_1_-DNA adduct formation in a positive linear relationship to dose [124]. This indicates that in rats, AFB_1_ does distribute and metabolize at human exposure relevant dosages. The limit of detection of analytical instruments range from 0.008 ng AFB_1_/mL of reconstituted sample on a HPLC-FLD or HPLC-MS/MS system [113,125] to 0.2 pg AFB_1_-lysine/mg albumin on a HPLC-MS/MS system [113,126]. A chronic, low dose rodent model of human exposure can recapitulate human dietary exposure in a detectable manner.

### 3.2. Accurate Immunological Data Generation

Inflammation status in the colon and vagina require specific biomarkers from locally acquired specimens. There is little correlation between the elevation of cytokines in the genital tract and that of blood plasma, indicating that cytokine concentrations at sites of HIV transmission are not related to systemic cytokine concentrations [42,127,128]. The detection of vaginal inflammation relies upon bio-fluid sampling, such as cervicovaginal lavage, for local cytokine analysis [127,129]. Specific, local markers of inflammation would produce more accurate inflammation status data at sites of HIV transmission than relying upon circulating cytokine detection.

Immune cells have location specific expression that can affect model study outcomes. The composition of intestine and cell-regulating cofactor expression changes along the gastrointestinal tract, the colon has unique antigen recognition and cytokine production relative to the small intestine [130,131]. Likewise, dendritic cells associated with the intestine are diverse in lineage, cell surface markers, and cytokine production [132]. Dendritic cells (DCs) transport HIV virions from the site of sexual contact to susceptible CD4^+^ T cells, known as trans-infection [133]. They can be native to a tissue or recruited from the periphery during inflammation [132]. It is unclear if recruited DCs can be converted by local tissue developmental factors to become intestine native-like DCs [132]. AFB_1_ treatment of CD11C^+^ myeloid dendritic cell lines upregulate production of pro-inflammatory cytokines IL-6 and IL-1β, and IL-6 and IL-1β upregulation is associated with HIV risk and advancement [45,51,52,134]. Research utilizing a variety of DCs relevant to colorectal, GI, and cervicovaginal tissues would help define the role of different DC populations in HIV infection, progression, and persistence [133]. Defining shared immunological pathways affected by HIV and inflammatory exposures, such as AFB_1_ ingestion, in appropriate cells, in the appropriate compartment, will increase accuracy and resolution.

## 4. Conclusions

There are several overlapping geo-local, tissue distribution, and metabolic pathway areas that AFB_1_ and HIV share, which we have highlighted in this review. Though the evidence is correlative, the overlap warrants further investigation into the mechanistic effects of AFB_1_ exposure on mucosal tissues and those impacts on HIV susceptibility. The current UNAIDS recommendation of combination HIV prevention relies on identifying and reducing multiple HIV risk factors, including behavioral, biomedical, and structural components [135]. There is increasing evidence that inflammation and mucosal barrier disrupting pollutants are risk factors for infectious diseases that enter through the mucosal barrier [32,136,137,138,139,140,141]. Investigating this pathway to increased HIV susceptibility potentially provides an easily modifiable component of combination HIV prevention.

There is a higher per-sex-act risk of HIV transmission in low- and middle-income countries, relative to high income countries, when known risk factors are accounted for [8]. This indicates that there are unaccounted risk factors plaguing high HIV case regions and that there is a geo-local component. Recently, the SARS-CoV-2 pandemic was examined in the context ofenvironmental pollutant co-exposure (airborne particulate matter, SARS-CoV-2, and disease outcome), aidedby pre-existing air quality monitoring infrastructure, showing a positive correlation [142]. Much of the evidence in this field, and in this review, is strongly correlative, with extrapolated clinical and molecular support that warrants further causative and mechanistic research. At this time, we do have validated tools to assess the potential linkage of HIVseroconversion and AFB_1_ exposure thourgh the endpoints of tissue morphology, cytokikne/chemokine regulation, AFB_1_ ADME, and HIV status/viral load. The barriers to eliminating or eradicating infectious diseases are many, leading to the need for advancing disease management strategies. Increasing resiliency to infectious disease by decreasing environmental pollutant induced mucosal barrier disruption and subsequent microbial translocation is a long overdue and logical focus for infectious disease research.

## Figures and Tables

**Figure 1 pathogens-10-01229-f001:**
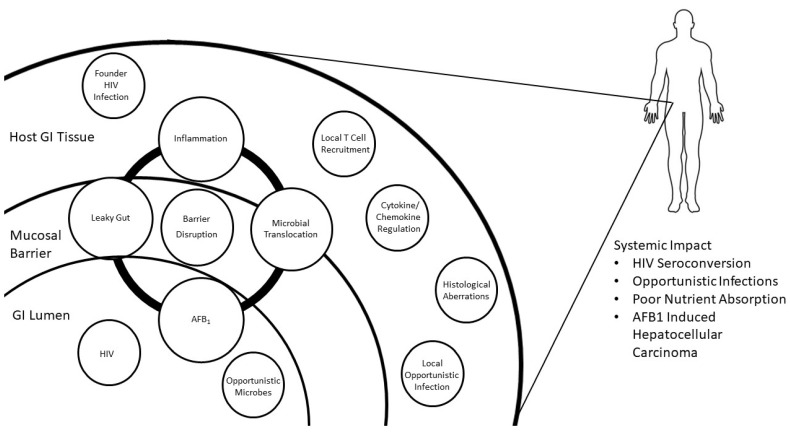
Mechanistic Etiology of AFB_1_ Induced HIV Susceptibility_._ The network of effects following colorectal exposure to HIV and AFB_1_. AFB_1_ is absorbed across the mucosal barrier of the gut lumen, generating inflammation and mucosal barrier disruption, leading to increased penetrance of microbes including HIV. AFB_1_ driven inflammation recruits an increased concentration of cells susceptible to HIV to the GI. The HIV pathway begins with viral delivery to the gut lumen, where it reaches the mucosal barrier and susceptible cells that are affected by AFB_1_ exposure.

**Table 1 pathogens-10-01229-t001:** Cytokines/Chemokine Regulation Associated With HIV in Sexual Tissues and Relevant AFB_1_ Regulation: Endpoints Under-Studied in Sexual Tissues After AFB_1_ Exposure.

HIV					AFB_1_ Exposure		
	Reg ^₸^.	Outcome	Tissue	Ref.	Reg ^₸^.	Model, Tissue	Ref.
IL-1α	U	risk factor	vaginal mucosa	[42]	U	rat, liver	[44]
IL-1β	U	risk factor	vaginal mucosa	[42]	U	dendritic cells	[45]
IL-2	U	susceptibility	resting T cells	[46,47]	D	chicken, GI	[48]
	U	replication	resting CD4^+^ T cells	[49]	D	rat, SMC *	[50]
	U	disrupted epithelial tight junctions/barrier	intestinal mucosa	[51]			
IL-4	U	susceptibility	resting T cells	[52,53]	U	rat, SMC *	[50]
IL-6	U U U U	incr. replication during infection activates replication disrupted epithelial tight junctions/barrier	vaginal mucosa intestinal cell model resting CD4^+^ T cells intestinal mucosa	[3,54,55] [51] [49,52] [51]	U D	dendritic cells chicken, ileum	[45] [56]
IL-7	U	incr. susceptibility	resting T cells	[52,53]			
IL-8	U	risk factor	vaginal mucosa	[42]	U	Jurkat T cells	[57]
IL-12	U U	during replication incr. local CD4^+^ T cells	vaginal mucosa intestinal mucosa	[54,55] [3] [46,52]			
IL-15	U	incr. susceptibility	Resting T cells	[52,53]			
IL-17	D	microbial translocation	intestinal mucosa	[52,58]	U D	mouse, liver chicken, GI	[59] [48]
IL-18	U U U U	replication ART failure incr. replication incr. local CD4^+^ T cells	vaginal mucosa blood plasma monocytes and T cells intestinal mucosa	[3,54,55] [52,60] [52,60] [46,52]			
IL-22	D	microbial translocation	intestinal mucosa	[52,58]			
TNF-α	U U U U U	during replication risk factor during infection activates replication disrupted epithelial tight junctions/barrier	vaginal mucosa vaginal mucosa intestinal cell model resting CD4^+^ T cells intestinal mucosa	[3,54,55] [42] [51] [49,52] [51]	U D	rat, serum chicken, GI	[61] [48,56]
IFN-Υ	U	during replication	vaginal mucosa	[3,54,55]	U U	pig, spleen mouse, liver	[47] [59]
MIP-1a	U	risk factor	vaginal mucosa	[42]			
MIP-1b	U	risk factor	vaginal mucosa	[42]			

^₸^ U/D = up/down regulation; * SMC = spleen mononuclear cells.

## Data Availability

Not applicable.

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
