# Peer review of "Environmental Pollutants, Mucosal Barriers, and Pathogen Susceptibility; The Case for Aflatoxin B1 as a Risk Factor for HIV Transmission and Pathogenesis"

_pathogens, 2021, doi:10.3390/pathogens10101229_

Round 1
Reviewer 1 Report
Hello, I would consider this manuscript for publication pending following minor changes that I think would improve the quality of the manuscript.
Section 2: title could be changed to reflect the main message of the section, for example it could be rephrased to " Effects of AFB1 exposure on HIV susceptibility" to identify the message of this section more clearly.
Figure 1 could be simplified to make the main point more obvious. Could the authors please divide the figure into multiple parts and lesser arrows?
The description of immune cell profile in HIV patients can be edited to make it more relevant in the context of exposure to AFB1 .
Reviewer 2 Report
Summary: Madeen and colleagues provide data in a lengthy review to support their hypothesis that environmental pollutants, that includes the carcinogen aflatoxin B1 (AFB1), are risk factors for the susceptibility and infectivity of sexually acquired HIV. Much of the review focuses on mucosal barriers and includes a large comprehensive table that summarizes cytokines and chemokines that are said to link HIV infection with AFB1 exposure. The authors conclude that the evidence presented suggests that AFB1 exposure is indeed a risk factor for HIV transmission.
Review: Although well written, this review suffers from many weaknesses that are summarized below:
- While presented as a review to support the premise that AFB1 is an environmental cofactor in the transmission of HIV, it is really masquerading as a vehicle to review mucosal barriers vis-a-vis HIV transmission and infection. The review is far too lengthy and should be greatly reduced in size to focus only on AFB1 and HIV transmission and infection.
- On the one hand, the reader is told that the review focuses on AFB1 exposure and sexual transmission of HIV (see Lines 21 and 22 of the Abstract), yet, on the other hand, large parts of the review discusses AFB1 and mucosal barriers of the gut (see Section 2.1.6). The review is schizophrenic and never really decide its ultimate focus and goals.
- The section on STIs (Section 2.1.3) has nothing to do with AFB1 exposure. Moreover, this section ignores the contribution of syphilis to HIV transmission because the primary chancre is noted to be a major portal of entry of HIV into the body. It would be also helpful to explain to the reader why gonorrhea, an STI without external lesions, contributes significantly to HIV transmission.
- Table 1 is the major weakness of the review (fatal flaw?). It is constructed apparently to provide a link between HIV infection and AFB1 exposure with the common thread being upregulation or downregulation of several listed cytokines and chemokines. This is very misleading. While true that the cytokines and chemokines listed are upregulated or downregulated during HIV infection as supported by the many references, where is the evidence that these altered patterns of cytokine and chemokine production during HIV infection are also altered in identical patterns after AFB1 exposure? This list could easily be applied subjectively to other environmental factors!
- Overall, the findings presented to link HIV transmission and infection with AFB1 exposure are circumstantial, a conclusion admitted by the authors in the Abstract (Line 27).
Reviewer 3 Report
The manuscript by Madlen et al. reviews a potential correlation between Aflatoxin B1 exposure and increased risk of getting infected and spreading HIV.
This review is largely speculative but makes an interesting case on a subject that warrants further investigation. Since a large proportion of the cited references is already secondary literature, I would suggest clearly indicating these references such that the reader can more readily judge the presented evidence.
Round 2
Reviewer 2 Report
The manuscript has undergone substantial revision in response to reviewers' comments and concerns. While evidence is still circumstantial, the authors have honestly stated this fact to the reader. The review may stimulate further investigations of AFB1 and HIV infection.